# Seasonal Influence of Biodiversity on Soil Respiration in a Temperate Forest

**DOI:** 10.3390/plants11233391

**Published:** 2022-12-05

**Authors:** Mengxu Zhang, Emma J. Sayer, Weidong Zhang, Ji Ye, Zuoqiang Yuan, Fei Lin, Zhanqing Hao, Shuai Fang, Zikun Mao, Jing Ren, Xugao Wang

**Affiliations:** 1CAS Key Laboratory of Forest Ecology and Management, Institute of Applied Ecology, Chinese Academy of Sciences, Shenyang 110016, China; 2University of Chinese Academy of Sciences, Beijing 100049, China; 3Key Laboratory of Terrestrial Ecosystem Carbon Neutrality, Shenyang 110016, China; 4Lancaster Environment Centre, Lancaster University, Lancaster LA1 4YQ, UK; 5Smithsonian Tropical Research Institute, Panama City 32402, Panama; 6School of Ecology and Environment, Northwestern Polytechnical University, Xi’an 710072, China

**Keywords:** soil fungal diversity, soil water content, soil CO_2_ efflux, spatial heterogeneity, tree diversity

## Abstract

Soil respiration in forests contributes to significant carbon dioxide emissions from terrestrial ecosystems but it varies both spatially and seasonally. Both abiotic and biotic factors influence soil respiration but their relative contribution to spatial and seasonal variability remains poorly understood, which leads to uncertainty in models of global C cycling and predictions of future climate change. Here, we hypothesize that tree diversity, soil diversity, and soil properties contribute to local-scale variability of soil respiration but their relative importance changes in different seasons. To test our hypothesis, we conducted seasonal soil respiration measurements along a local-scale environmental gradient in a temperate forest in Northeast China, analyzed spatial variability of soil respiration and tested the relationships between soil respiration and a variety of abiotic and biotic factors including topography, soil chemical properties, and plant and soil diversity. We found that soil respiration varied substantially across the study site, with spatial coefficients of variation (CV) of 29.1%, 27.3% and 30.8% in spring, summer, and autumn, respectively. Soil respiration was consistently lower at high soil water content, but the influence of other factors was seasonal. In spring, soil respiration increased with tree diversity and biomass but decreased with soil fungal diversity. In summer, soil respiration increased with soil temperature, whereas in autumn, soil respiration increased with tree diversity but decreased with increasing soil nutrient content. However, soil nutrient content indirectly enhanced soil respiration via its effect on tree diversity across seasons, and forest stand structure indirectly enhanced soil respiration via tree diversity in spring. Our results highlight that substantial differences in soil respiration at local scales was jointly explained by soil properties (soil water content and soil nutrients), tree diversity, and soil fungal diversity but the relative importance of these drivers varied seasonally in our temperate forest.

## 1. Introduction

Forests store ~45% of terrestrial biomass carbon and play a crucially important role in the global carbon cycle [1]. Forests not only sequester and store carbon in biomasses and soils, but they release large amounts of carbon dioxide (CO_2_) back into the atmosphere through respiration from plants and soil. Quantifying soil respiration is particularly important for accurate predictions of global C cycling because it represents 60–80% of photosynthetic production (80–98 Pg C yr^−1^) and accounts for 40–90% of the global CO_2_ emissions from terrestrial ecosystems [2]. Indeed, forest soil respiration releases ten times more CO_2_ into the atmosphere than current human fossil fuel consumption [3]. However, there is considerable uncertainty around estimates of soil respiration from forest ecosystems due to high spatial and temporal variation. Most studies of soil respiration in forests have focused on temporal variation [4], which is often quantified using continuous automated measurements at a few sampling points within a given forest. However, soil respiration can vary two- to eight-fold within tens of meters [5,6], making it difficult to fully capture the spatial variability of soil respiration without continuous large-scale observations. Current knowledge of the underlying mechanisms of spatial variability of soil respiration in forests is relatively limited, which creates great uncertainty when estimates of soil respiration are upscaled to the ecosystem, regional, or global scale [7].

The high temporal and spatial variability of soil respiration results from numerous individual organisms and processes that contribute to total CO_2_ efflux from the soil. Soil respiration is often partitioned into heterotrophic respiration from microorganisms in the bulk soil and rhizosphere respiration, comprising the CO_2_ efflux from plant roots, their microbial symbionts, and other rhizosphere microorganisms [8]. These two major components of soil respiration each involve various biological processes from numerous organisms, which respond differently to environmental conditions, such as topography, microclimates, and soil elemental concentrations. Soil elemental concentrations influence soil respiration by determining the availability of nutrients and carbon for plant and microbial metabolism [9]. Soil respiration is also strongly regulated by microclimates, including soil temperature and water content. Heterotrophic respiration generally increases exponentially with temperature within typical soil temperature ranges due to greater microbial enzyme activity, greater substrate affinity, and enhanced substrate diffusion rates [10]. By contrast, soil respiration is often highest at intermediate soil water content because CO_2_ efflux is limited by substrate transport as well as microbial physiology and plant activity at low soil moisture levels, but limited by oxygen availability at high soil moisture levels [11]. Many forests have high soil water content, which can result in anaerobic conditions, thereby reducing soil CO_2_ emissions from roots and soil microorganisms [12]. However, topography can modify the hydrological conditions and other biophysical variables, which complicates assessments of the spatial and temporal variation of soil respiration [13]. Importantly, as plant growth and microbial activities are influenced by seasonal temperature and precipitation, the drivers of soil respiration can also vary across seasons [14,15]. Indeed, compared with abiotic factors, the relative contribution of biotic factors, such as plants and soil microbes, to soil variation could be more variable in both space and time.

Plant and microbial communities play decisive roles in regulating soil respiration since they are the principal circular pathways through which carbon enters the soil and is released back into the atmosphere. Plant diversity increases soil autotrophic respiration by enhancing metabolic rates and fine root biomass [16], but it also stimulates soil heterotrophic respiration due to a greater amount and variety of carbon and nutrient resources available for soil microorganisms [17]. In addition, stand structural complexity (e.g., individual tree size variation) could indirectly affect soil respiration by altering understory light environments, understory plant diversity [18], soil temperature variability [19], and soil microbial activity [20]. However, the influence of plant diversity on soil respiration is likely to vary substantially in time, as root growth and plant litter inputs are inextricably linked to plant growth and therefore often show seasonal patterns [21]. Besides differences in root respiration, plant diversity primarily influences soil respiration via the quality and quantity of plant inputs available to microbes [22]. It is highly likely that soil microbial diversity plays critical roles in soil C cycling in terrestrial ecosystems. However, empirical evidence demonstrating the role of microbial community composition in driving C fluxes such as soil respiration is limited, especially in natural forests. Some experimental evidence has shown that loss of microbial diversity led to higher rates of soil microbial respiration [23]. High soil microbial diversity not only limits community activity by increasing interspecific competition [24], but diverse communities can also contain species that contribute less to heterotrophic respiration [25]. Importantly, soil microbial diversity and community composition are closely associated with plant diversity but also display seasonal patterns that follow changes in soil temperature and soil water content [26]. However, to date, we do not know how above- and belowground diversity collectively shape seasonal differences in soil respiration at local scales.

Large forest dynamics plots provide a great opportunity to understand how plant attributes (e.g., plant biomass, species diversity, and stand structure) and soil diversity regulate soil respiration in forests while accounting for environmental conditions (microclimate and soil nutrients). Here, we conducted field measurements of soil respiration at 150 sampling points uniformly covering a 25 ha permanent temperate mixed forest plot in Northeast China in different seasons (spring, summer, and autumn). We hypothesize that: (1) The relative importance of biotic and abiotic factors in determining local variation in soil respiration varies among seasons; and (2) Above- and belowground species diversity and soil properties jointly regulate local variation in soil respiration, whereby soil respiration would increase with plant diversity but decline with increasing soil diversity.

## 2. Results

### 2.1. Seasonal and Spatial Variability of Soil Respiration

Soil respiration rates differed among seasons, with the highest rates in summer (4.82 ± 1.32 μmol CO_2_ m^−2^ s^−1^), followed by spring (3.21 ± 0.93 μmol CO_2_ m^−2^ s^−1^), and the lowest in autumn (2.25 ± 0.69 μmol CO_2_ m^−2^ s^−1^). In addition, the range of soil respiration rates was also largest in summer (1.65~8.34 μmol CO_2_ m^−2^ s^−1^), somewhat smaller in spring (1.34~7.16 μmol CO_2_ m^−2^ s^−1^), and smallest in autumn, with much lower minimum values (0.69~4.95 μmol CO_2_ m^−2^ s^−1^; Table 1). Nonetheless, spatial variability of soil respiration was similarly high in all seasons, with a spatial variation coefficient (CV) of 29.1%, 27.3%, and 30.8% in spring, summer, and autumn, respectively.

### 2.2. Abiotic and Biotic Factors Influencing Spatial Variation of Soil Respiration

Of all measured potential predictors of soil respiration, seasonal variation was highest for soil temperature, with values of 12.3 ± 0.7 °C in spring, 16.8 ± 0.8 °C in summer, and 12.3 ± 1.4 °C in autumn. However, the spatial CV for soil temperature was low (5.7%, 4.5%, and 11.0%, in spring, summer, and autumn, respectively). By contrast, soil water content varied less across seasons, with values of 39.8 ± 8.0% in spring, 36.3 ± 8.9% in summer, and 41.9 ± 7.3% in autumn, but the spatial CV of soil water content was high in each season (20.1%, 24.6%, and 17.3%, in spring, summer, and autumn, respectively). Of the tree community parameters, spatial variation in thetree basal area was highest, with a spatial CV of 29.8%, compared to 21.9% for tree species richness and 15.7% for tree size variation. Finally, of the soil diversity parameters, bacteria had higher diversity (6.54 ± 0.12) but lower spatial variation (1.8%) than fungi (3.84 ± 0.53; CV 13.7%) or nematodes (5.24 ± 0.82; CV 15.6%).

Multiple linear regression showed that soil respiration was related to distinct abiotic and biotic factors depending on the season (Figure 1a–c). Soil respiration was not related to topography in any season but declined strongly with increasing water content in all seasons, especially in summer (β = −0.22, *p* < 0.01, Appendix A). Soil respiration increased with temperature only in summer (β = 0.05, *p* < 0.05; Figure 1), and declined with increasing total soil elements in autumn (β = −0.09, *p* = 0.01).

Overall, soil respiration increased with tree biomass and diversity, but the strength of the relationship varied among seasons. In spring, soil respiration was highest in subplots with high tree basal area (β = 0.06, *p* < 0.05) and tree diversity (β = 0.09, *p* < 0.01) but there was no relationship between soil respiration and basal area in summer or autumn. Soil respiration generally declined with increasing soil diversity, but the relationship was only significant for fungal diversity in spring (β = −0.05, *p* < 0.05). In summary, low rates of soil respiration were associated with increasing soil moisture, total soil elements, and fungal diversity, whereas high respiration rates were associated with increasing soil temperature, and high tree biomass and diversity, but the strength of the relationships varied markedly among seasons.

### 2.3. Interactive Effects of Main Factors on Soil Respiration

The direct paths in the SEMs conformed to the results of the multiple linear regressions. Among the measured abiotic factors, soil water content had the greatest direct effect, as lower rates of soil respiration were associated with high soil water content in all seasons (Figure 2a–c). In autumn, soil respiration was also lower in subplots with high concentrations of total soil elements (Figure 2c) but none of the other abiotic factors were associated with local-scale differences in soil respiration rates. Of the measured biotic factors, soil respiration was most strongly associated with tree diversity, with higher rates of soil respiration in subplots with high tree diversity in all seasons (Figure 3a,b). In spring, subplots with greater tree biomass had higher rates of soil respiration (Figure 3a), but subplots with high soil fungal diversity had lower rates of soil respiration.

Indirect paths in the SEMs revealed that higher rates of respiration were associated with total soil elements in all seasons via tree diversity (Appendix A). In summer and autumn, high soil water content was also associated with reduced tree biomass and stand structural complexity (Figure 2b,c). Although stand structure had no direct effect on soil respiration, it was associated with higher respiration rates via tree diversity in spring (Figure 2a). Overall, soil water content and tree diversity were identified as the two strongest predictors of spatial variation in soil respiration (Figure 3). The relative contribution of soil water content to variability in soil respiration in spring, summer, and autumn was 17.2%, 60.5%, and 28.4%, respectively, (Figure 3a), whereas tree diversity contributed 27.4.1%, 18.7%, and 23.9%.

## 3. Discussion

Our study evaluated the relative importance of plant and soil diversity in explaining local-scale variability in soil respiration in different seasons in a temperate forest. We demonstrate that high rates of soil respiration were associated with tree community characteristics (i.e., tree biomass, stand structural complexity, and tree species richness), whereas lower respiration rates were associated with belowground abiotic (soil water content and total soil elements) and biotic factors (soil fungal diversity). Importantly, the strength of these above- and below-ground factors as predictors of soil respiration varied greatly among seasons. How soil respiration might be influenced by complex relationships among abiotic conditions and multiple trophic groups throughout the year has rarely been investigated [27]. Thus, our work adds considerably to our understanding of local-scale variability in soil respiration.

### 3.1. Seasonal and Spatial Variability of Soil Respiration

The clear seasonal dynamics of soil respiration, with a peak during the summer, is characteristic of temperate forest systems, where high temperatures in summer stimulate the growth and metabolic rates of trees and microbes [28]. Compared with soil water content, the seasonal dynamics in soil respiration was predominantly determined by soil temperature (Appendix A), probably because temperature substantially influences plant phenology and soil microbial seasonal activity in temperate forests [29].

Soil respiration was highly variable across the local spatial scales measured in our study, with spatial CVs ranging from 27% to 31%. The spatial variability in our forest fell within the wide ranges previously recorded in Chinese forests (17–62.6%; [30]). Other studies in similar forests have observed both higher and lower spatial variability [31,32], demonstrating the importance of sampling design to gain accurate estimates of soil respiration in forests [7]. High spatial variability in soil respiration can result from excessively dense sampling. For example, Shi et al. [32] found that 87–91% of the spatial variance was explained by an autocorrelation over a range of 15 to 23 m. By contrast, the minimum sampling distance in our study was >28 m, so we only detected a weak spatial autocorrelation of soil respiration in spring.

### 3.2. Relationships between Abiotic Factors and Soil Respiration

In support of our first hypothesis, soil water content played the most important role in inhibiting soil respiration during the growing season, which was probably due to high rainfall and the high water holding capacity of the forest floor. Previous studies have revealed a threshold value of soil water content (approximately 20%), at which the relationship between soil respiration and soil moisture changes [33]. At soil water content above the threshold value, soil respiration is limited due to low CO_2_ transport, oxygen (O_2_) availability, and thus reduced biological activity. By contrast, soil water contents below the threshold promote root and soil microbial respiration through the diffusion of soluble substrates in an aerobic environment [33]. In our study, 98% of the measurements showed a soil water content greater than 20%, which explains why soil respiration rates declined with increasing soil water content. The limiting effect of high soil water content was strongest in summer, due to the high autotrophic and heterotrophic respiration and thus greater requirement of oxygen availability and soil porosity [34]. Indeed, soil water content in summer explained a much higher proportion of the variability in soil respiration (60.5%) than any other abiotic or biotic factors.

Surprisingly, soil properties, including organic matter and total and extractable nitrogen, phosphorus, and potassium, were not strong predictors of soil respiration in our study. In general, high-nutrient substrates should increase soil heterotrophic respiration by enhancing microbial biomass and fungal abundance [9]. However, late-successional forests such as ours often generally have high soil nutrient concentrations, and nutrient availability is therefore less likely to limit soil respiration [35]. Instead, the indirect association between total soil elements and soil respiration in our SEMs (Figure 2) suggests that nutrient availability increased soil respiration by promoting plant diversity [36].

### 3.3. Contrasting Relationships between Soil Respiration and Above- or Belowground Diversity

Higher rates of soil respiration at sites with greater tree species richness have also been observed in other ecosystems and on a global scale [37]. Plant species richness is usually associated with greater chemical diversity of litter and root exudates [16], which stimulates soil heterotrophic respiration by providing a greater range of carbon and nutrient resources to soil microorganisms [38]. Although we found no relationship between tree species richness and soil diversity (Appendix A), previous work at the same study site demonstrated that soil microbial diversity was linked to the functional diversity of the tree community, rather than species diversity [39]. Thus, the strong role of tree species diversity in explaining spatial variability in soil respiration is likely a combination of species differences in the autotrophic component of soil respiration, as well as differences in tree litter quality and quantity influencing decomposer organisms and heterotrophic respiration.

The stronger relationship between soil respiration and fungal diversity compared to soil bacteria and nematodes, is consistent with previous studies [40], and is often explained by the key role of fungi in decomposing recalcitrant plant materials [41]. However, in our study, soil respiration in spring declined with increasing fungal diversity (Figure 1), supporting our second hypothesis. The negative relationship between soil respiration and soil fungal diversity could be the result of higher carbon use efficiency by a diverse fungal community benefitting from distinct resource niches [42], which does not necessarily increase overall community activity and heterotrophic respiration [43]. Alternatively, differences in the abundances and diversity of distinct fungal functional guilds could influence the relationship between soil respiration and overall fungal diversity. In our study, ectomycorrhizal (EcM) fungi accounted for 40.8% of the total fungal abundance, but only contributed 14.8% to fungal species richness (Appendix A). Furthermore, ectomycorrhizal fungi account for a large proportion of total soil respiration (15–26%; [44,45]), so the dominance of ectomycorrhizal fungi could explain why soil relationship rates were highest in subplots with low fungal diversity.

Although greater tree biomass is thought to stimulate soil microbial growth and activity by increasing plant inputs, such as litter and root exudates to the soil [46], we found only a weak relationship between basal area, as a proxy of tree biomass, and soil respiration in spring (Figure 1). It is possible that the relationship between soil respiration and tree biomass in summer and autumn was obscured by the overriding influence of soil water content [47], when soil microbial activity or heterotrophic respiration was probably limited by anoxic soil conditions. Although soil respiration was also not directly related to stand structure (represented by tree size variation), our SEMs revealed that a more complex stand structure was associated with higher respiration rates via tree diversity (Figure 2). Structurally complex forests have a greater range of niches and more diverse light conditions, which increase understory plant abundance and richness, thereby enhancing autotrophic and heterotrophic respiration [48].

## 4. Materials and Methods

### 4.1. Study Site Description and Experimental Design

The study site was located in the Changbai Nature Reserve in Northeast China, extending from 41°42′ to 42°26′ N and 127°42′ to 128°17′ E. As one of the largest biosphere reserves in China, the Changbai Nature Reserve was established in 1960 and joined the World Biosphere Reserve Network in 1980. Mean annual precipitation is approximately 700 mm and most rainfall occurs from June to August (450–500 mm). Mean annual temperature is 2.8 °C, with monthly means of −13.7 and 19.6 °C in January and July, respectively [49]. Previous work in the study forest indicates a total annual soil respiration of 1017 g m^−2^, accounting for 76% of ecosystem respiration [50].

Our study was conducted in the 25 ha Changbaishan (CBS) Forest Dynamics Plot (FDP), which is one of the sites in the worldwide CTFS-ForestGEO forest monitoring network (http://www.forestgeo.si.edu, accessed on 5 January 2021). All free-standing individual woody plants with a diameter at breast height (DBH) ≥1 cm were mapped, measured, and identified to species in 2004, 2009, 2014, and 2019. Based on the first census data in 2004, there are 38,902 individuals belonging to 52 species, 32 genera, and 18 families [51]. The 25 ha plot is dominated by the late-successional deciduous broadleaved Korean pine (*Pinus koraiensis*) mixed forest with common tree species including *P. koraiensis*, *Tilia amurensis*, *Quercus mongolica*, *Fraxinus mandshurica*, *Ulmus japonica*, and *Acer mono*. Following a standard field protocol [52], the 25 ha plot was divided into 625 subplots (20 m × 20 m). We measured three topographical variables (elevation, slope, and convexity) for each subplot following Harms et al. [53]. Elevation of each subplot was estimated from the mean elevation at each corner. The slope was defined as the mean angular deviation from horizontal of each of the four triangular planes formed by connecting three subplot corners. Subplot convexity was calculated as the elevation of the subplot minus the mean elevation of the eight surrounding subplots; for edge subplots, convexity was calculated as the elevation of the central point minus the mean of the four corners.

To capture the spatial variability of soil respiration, 150 sampling points were established as evenly as possibly across the 25 ha plot (Figure 4). In April 2020, permanent soil collars made of polyvinyl chloride (20 cm diameter and 10 cm height) were inserted 4 cm into the soil at each sampling point and left in situ throughout the soil respiration measurements. Soil collars set at this depth were stable and caused minimal disturbance to shallow fine roots. To avoid the confounding effects of above-ground plants on soil respiration, we removed small living plants growing inside the collars one day before each measurement.

### 4.2. Measurements of Soil Respiration and Soil Microclimates

Soil respiration, temperature, and water content were measured seven times during the growing season of 2020 (May to October): twice in spring (May–June), three times in summer (July–August), and twice in autumn (September–October). Soil respiration was measured using an automated soil CO_2_ flux system, which consisted of a dynamic soil chamber (3140 cm^3^ volume) attached to an infrared gas analyzer (Li-8100A, Li-COR Inc., Lincoln, NE, USA). Each measurement lasted 90 s, recording CO_2_ and water concentrations and air temperature inside the chamber every second. Soil temperature and volumetric water content at 0–5 cm soil depth were determined adjacent to each collar using a type E thermocouple (8100-201, Li-COR Inc., Lincoln, NE, USA) and a ML2x soil moisture probe (Li-COR Inc., Lincoln, NE, USA), respectively. During each sampling campaign, measurements took two days to complete. As diel variation in soil CO_2_ efflux is low in heavily shaded forested areas [54], all measurements were performed in a random order between 10 a.m. and 4 p.m. on each measurement day. We did not measure soil respiration in winter, as low temperatures strongly constrain microbial activity.

### 4.3. Sampling and Analysis of Soil Physicochemical Properties

To represent the average soil properties associated with plant communities, we randomly selected two soil sampling sites at the midpoints between the central point and the four corners in the above-mentioned 150 subplots (20 m × 20 m). At each site, we took five soil cores (3.8 cm in diameter and 10 cm in depth) at random locations near the 150 sampling points using a soil auger after removing the litter layer from the ground surface. Subplots located at the edge of 25 ha plot were not included to eliminate potential edge effects resulting in a total of 120 quadrats. We mixed the cores to create one composite sample per measurement point. Each soil sample was then divided into two parts after sieving the sample through a 2 mm mesh to remove the roots and stones: one for soil nutrient analysis and the other for soil diversity measurement (i.e., bacteria, fungi, and nematodes). All processing was completed within 12 h of collection, and the subsamples for soil diversity analysis were stored at −80 °C.

We measured eight soil nutrient variables known to influence soil respiration: soil pH, organic matter (SOM), extractable nitrogen (N_EX_), extractable phosphorus (P_EX_), extractable potassium (K_EX_), total nitrogen (TN), total phosphorus (TP), and total potassium (TK). Following Lu [55], soil pH was determined in water (1:1 soil:solution ratio) using a glass electrode, SOM was determined colorimetrically following dichromate oxidation, N_EX_ was determined by extraction in 1 mol NaOH L^−1^ and subsequent titration with 0.01 mL·L^−1^ sulfuric acid, TN was estimated colorimetrically after KCl extraction, using the Kjeldahl method, and P_EX_ and TP were determined by molybdate colorimetry, after Mehlich 3 extraction or digestion in H_2_SO_4_–HClO_4_, respectively; K_EX_ and TK were determined by atomic absorption spectrometry after extraction with 1 mol L^−1^ NH_4_Ac or digestion in hydrofluoric acid, respectively.

### 4.4. Soil Diversity

Soil bacterial and fungal diversity and community composition were determined by sequencing on an Illumina Miseq platform (Illumina, San Diego, CA, USA). Soil genomic DNA was isolated from 0.25 g of fresh soil using the MoBio PowerSoil^®^ DNA Isolation extraction kit according to the manufacturer’s instructions (MoBio Laboratories Inc., Carlsbad, CA, USA). The quality of the DNA was assessed based on 260/280nm and 260/230 nm absorbance ratios obtained using a Spectrophotometer (NanoDrop Technologies Inc., Wilmington, DE, USA). Extracted DNA samples were stored at −20 °C until further use.

The universal bacterial V4~V5 region of 16S rRNA gene was amplified by using primers 515 F (5′-GTGCCAGCMGCCGCGG-3′) and 907 R (5′-CCGTCAATTCMTTTRAGTTT-3′) [56]. The fungal ITS sequence of 18S rRNA genes was amplified using primers ITS_1737F (5′-GGAAGTAA AAGTCGTAACAAGG-3′) and ITS_2043R (5′-ATGCAGGCTGCGTTCTTCA TCGATGC-3′) [57]. Amplification by polymerase chain reaction (PCR) was conducted in triplicate using a 20 µL mixture containing 4 µL of 5× FastPfu Buffer, 2 µL of 2.5 mM dNTPs, 0.8 µL of each primer (5 µM), 0.4 µL of FastPfu Polymerase, and 10 ng of template DNA was performed. The PCR analyses were carried out on a Gene Amp PCR-System 9700 (Applied Biosystems, Foster City, CA, USA) using thermal cycling conditions, as follows: initial denaturation step at 95 °C for 3 min, followed by 27 (16S rRNA) or 35 (ITS) cycles at 95 °C for 30 s, annealing at 55 °C for 30 s and extension72 °C for 45 s, and a final extension at 72 °C for 10 min. The PCR products were sequenced using 300PE MiSeq (Illumina, San Diego, CA, USA), at the Shanghai Majorbio Bio-pharm Biotechnology Co., Ltd. (Shanghai, China).

Obtained DNA sequences were processed using the QIIME 2 software [58], discarding sequences shorter than 200 bp with a mean quality score <25 and ambiguous bases. All quality-filtered sequences were clustered into Operational Taxonomic Units (OTUs) with a 97% identity threshold using UPARSE version 7.1 (http://drive5.com/uparse/, accessed on 14 May 2021). Chimeras were filtered using the Ribosomal Database Project (RDP) and UCHIME [59]. Saprotrophic vs. ectomycorrhizal fungal functional guilds were identified according to Yao et al. [60].

Nematodes were extracted from 200 g of fresh soil by an updated cotton-wool filter method [61]. For each sample, the first nematodes encountered on the slides were identified at genus level at 100× magnification under an inverted microscope. The nematodes were assigned to four trophic groups (bacterivores, fungivores, omnivores-predators, and plant parasites) according to Yeates et al. [62]. The abundance of nematodes was calculated as the number of individuals per 100 g dry soil. The detailed procedure for soil nematode extraction and identification has been described by Guan et al. [63]. Soil diversity in each subplot was represented by species richness calculated using the Shannon–Wiener index for soil fungi (Fungi_SWI_), bacteria (Bacteria_SWI_), and species richness of nematodes (Nematode_SR_).

### 4.5. Plant Community Characteristics

We described plant community characteristics using the latest census data of the plot in 2019, which measured and identified all woody plants (henceforth ‘trees’) with a diameter at breast height (DBH) >1 cm. We used species richness to represent tree diversity (Tree_SR_), basal area to represent biomass (Tree_BA_), and the coefficient of variation for DBH as a measure of tree size variation (Tree_SV_), which is a proxy of stand structure [64]. To determine the spatial influence of the tree community on soil respiration, we calculated tree community characteristics for a 5 m, 10 m, and 15 m radius around each sampling point. As the tree community characteristics within a 10 m radius showed the strongest correlation with soil respiration (Appendix A), we used the tree community characteristics within a 10 m radius for all subsequent analyses.

### 4.6. Statistical Analyses

All analyses were conducted in R 4.1.3 (R Development Core Team, Vienna, Austria, 2022), using the vegan package [65] to calculate tree species diversity and soil diversity indices, the MuMIn [66] and lavaan [67] packages for model averaging and structural equation models, respectively, and the spdep package [68] for Moran’s *I* test.

To assess spatial autocorrelation of soil respiration among sampling plots, we conducted Moran’s *I* test [69] in each season (spring, summer and autumn). We did not find any significant spatial autocorrelation in summer and autumn, and found a slight positive spatial autocorrelation in spring (Appendix A).

The eight measured soil properties (soil pH, organic matter content, N_EX_, P_EX_, K_EX_, TN, TP, and TK) were reduced to a set of orthogonal variables using principal component analysis (PCA). The first axis of the PCA (soil PCA1) explained 39% of the total variation and was positively correlated with total soil elements, whereas the second axis (soil PCA2) explained 20% and was positively correlated with extractable soil elements (Appendix A). Thus, both axes represented a soil fertility gradient from infertile to fertile soils (Appendix A) and were used in subsequent analyses to represent total soil elements (soil PCA1) and extractable soil nutrients (soil PCA2)

To examine the effects of topography, soil water content, soil temperature, soil properties, and tree or soil biodiversity on soil respiration, we performed multiple linear regressions models for soil respiration in each season. To avoid multicollinearity problems, we first assessed correlations between pairs of predictors within each group of variables (i.e., tree diversity, soil diversity, soil properties, topography) and excluded one predictor per pair if r > 0.60 (Appendix A); in each case, we retained the predictor that had greater explanatory power for variation in soil respiration. Using this approach, we included two parameters for tree communities (basal area and diversity), one for stand structure (tree size vatiation), three for soil diversity (Shannon’s diversity for fungi and bacteria, and species richness for nematodes), three parameters for topography (elevation, slope, and convexity), and four for soil properties (soil water content, soil temperature, total soil elements, and extractable soil nutrients), as predictors of soil respiration in the multiple linear regression models. Then, for each predictor, we performed all subsets regression analysis and selected the optimal model based on the lowest Akaike Information Criterion adjusted for small sample sizes (AICc). However, if the difference in AICc between models was <2 units, we obtained the standardized regression coefficient (*β*) of each soil respiration predictor using model averaging (Appendix A).

To identify the potential direct and indirect pathways and relative contributions through which soil properties and tree and soil communities regulate soil respiration, we built structural equation models (SEMs) following an initial conceptual model (Appendix A). In the SEMs, we used the significant predictors obtained from the optimal regression models: tree biomass, tree diversity, stand structure, and fungal diversity, as well as soil water content and total soil elements. The model fit to the data was evaluated using a Chi-square test (*p* > 0.05 indicates that the model is accepted), Bentler’s comparative fit index (CFI close to 1 indicates perfect model), and the standardized root mean square residual (SRMR < 0.08 indicates the most appropriate model) [67].

The indirect effect of each predictor in the final SEMs was calculated through the interaction of the standardized direct effect of a given predictor on a mediator with the direct effect of the mediator on the response variable. More specifically, the total indirect effect was calculated by multiplying the standardized direct effects of a given predictor on soil respiration via mediators for each relevant path, and then we added all indirect effects of the predictor to quantify the total indirect effect. To quantify the relative contribution of different predictors to soil respiration, we calculated the relative importance for each predictor of soil respiration using the ratio between the total effect of a given predictor and the sum of the absolute value of total effects of all predictors (Appendix A) [70].

## 5. Conclusions

Our work demonstrates high spatial variability of soil respiration on a local scale in a temperate forest, but the influence of biotic and abiotic factors on soil respiration changed during the growing season. Our findings enhance our understanding of spatial and temporal variability of soil respiration and can thus improve predictions of soil carbon flux in temperate forests. However, although local-scale differences in soil respiration could be partially attributed to differences in soil water content and tree or soil fungal diversity, a large proportion of variation remains unexplained (75%, 50%, and 79% in spring, summer, and autumn, respectively). Thus, to further reduce uncertainty in estimating soil respiration, more future studies should consider: (a) distinguishing between soil autotrophic and heterotrophic respiration; (b) assessing how plant functional diversity influences autotrophic and heterotrophic respiration; and (c) differences in soil microbial functional groups and their contribution to heterotrophic soil respiration.

## Figures and Tables

**Figure 1 plants-11-03391-f001:**
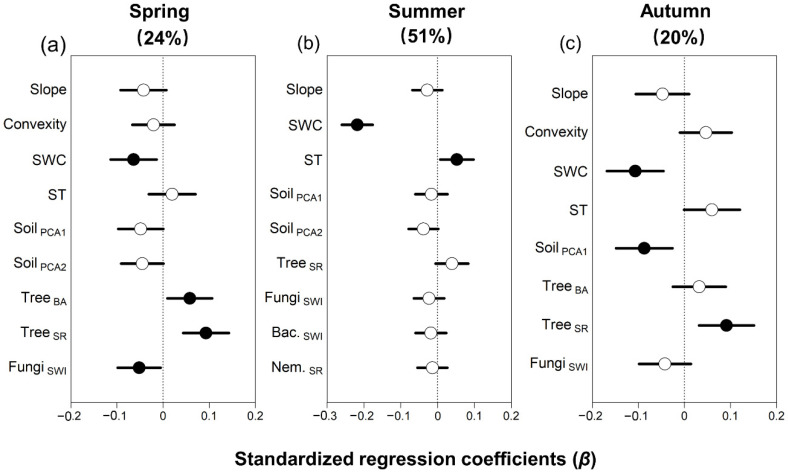
Standardized regression coefficients (*β*) of the biotic and abiotic predictors of soil respiration across a 25 ha forest dynamics plot in Northeast China, derived from multiple linear regression models for each season. The total R^2^ is given in parentheses for each model (See also Appendix A). Closed circles indicate significant relationships with soil respiration at *p* < 0.05, and lines indicate standard errors. All abbreviations follow Table 1.

**Figure 2 plants-11-03391-f002:**
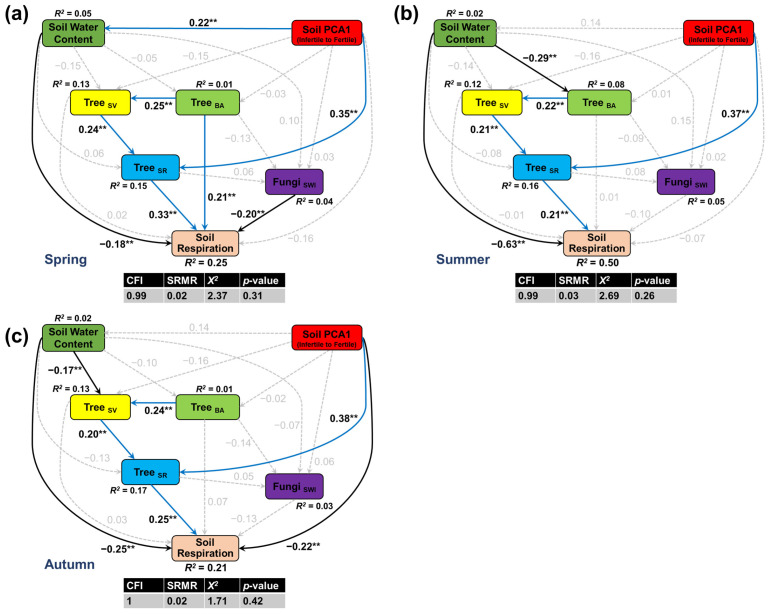
Structural equation models showing differences in soil respiration in spring (**a**), summer (**b**), and autumn (**c**) associated with soil moisture (soil water content), total soil elements (soil PCA1), plant biomass (Tree_BA_), stand structural complexity (Tree_SV_), plant diversity (Tree_SR_), and soil fungal diversity (Fungi_SWI_). Blue and black solid arrows represent significant positive or negative paths, respectively, at *p* < 0.05 (** *p* < 0.05); grey dashed arrows indicate non-significant paths. Values adjacent to arrows represent standardized coefficients (see Appendix A). All abbreviations follow Table 1.

**Figure 3 plants-11-03391-f003:**
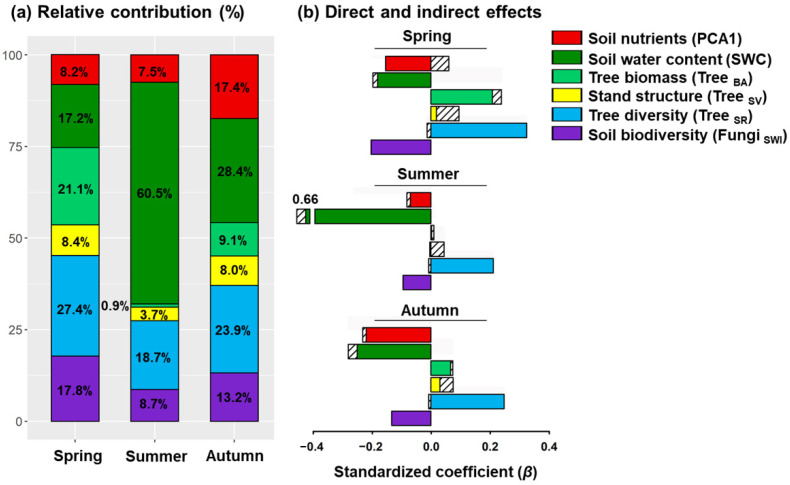
Relative contributions (**a**), and direct and indirect effects (**b**) of significant predictors to soil respiration rates in different seasons. Solid colours represent direct effects derived from structural equation models, whereas striped bars represent indirect effects (see also Appendix A).

**Figure 4 plants-11-03391-f004:**
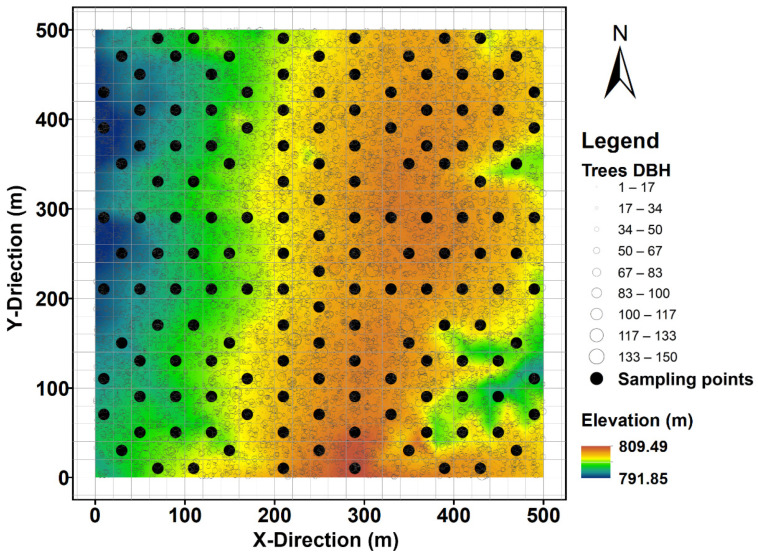
Contour map of the 25 ha Changbaishan Forest Dynamics Plot in Northeast China, showing the sampling points of soil respiration as solid circles and the positions of all living trees as open circles, whereby symbol sizes indicate tree diameter at breast height (DBH). Colours indicate differences in elevation.

**Table 1 plants-11-03391-t001:** Descriptive statistics for the soil respiration, above- and below-ground community structure and soil properties in a 25 ha temperate forest dynamics plot in Northeast China, showing maximum (Max.), minimum (Min.), and median (Med.) values as well as standard deviations (SD) and the coefficient of variation (CV) for *n* = 120 subplots. Rs is the mean soil respiration rate, ST is mean soil temperature, and SWC is mean soil water content, given for spring, summer, and autumn. Soil PCA1 and Soil PCA2 are ordination axes representing eight indexes of total soil elements or extractable nutrients and pH, respectively (Appendix A); Tree_BA_ is the total basal area of aboveground plants; Tree_SV_ is tree size variation; Tree_SR_ is the species richness of the plant community; Bacteria_SWI_ is the Shannon–Wiener index for soil bacteria; Fungi_SWI_ is the Shannon–Wiener index for soil fungi; Nematode_SR_ is the species richness of nematodes.

Variable	Mean	Max.	Median	Min.	SD	CV (%)
Rs_Spring_ (μmol CO_2_ m^−2^ s^−1^)	3.21	7.16	3.08	1.34	0.93	29.1
Rs_Summer_ (μmol CO_2_ m^−2^ s^−1^)	4.82	8.34	4.81	1.65	1.32	27.3
Rs_Autumn_ (μmol CO_2_ m^−2^ s^−1^)	2.25	4.95	2.26	0.67	0.69	30.8
ST_Spring_ (°C)	12.3	14.7	12.3	11.0	0.70	5.7
ST_Summer_ (°C)	16.8	18.2	16.8	15.4	0.75	4.5
ST_Autumn_ (°C)	12.3	14.8	12.3	8.6	1.35	11.0
SWC_Spring_ (%)	39.79	53.40	40.33	13.15	7.99	20.1
SWC_Summer_ (%)	36.31	52.63	35.07	11.53	8.93	24.6
SWC_Autumn_ (%)	41.88	52.95	44.11	20.85	7.25	17.3
Soil PCA1	0	5.43	−0.24	−4.49	1.77	-
Soil PCA2	0	3.41	−0.08	−4.61	1.26	-
Tree_BA_ (m^2^)	1.73	2.88	1.72	0.33	0.52	29.8
Tree_SV_	1.66	2.26	1.65	1.11	0.26	15.7
Tree_SR_	11	20	11	5	2.41	21.9
Bacteria_SWI_	6.54	6.82	6.55	6.27	0.12	1.8
Fungi_SWI_	3.84	5.26	3.91	2.40	0.53	13.7
Nematode_SR_	5.24	7.42	5.22	3.23	0.82	15.6
Elevation (m)	803.5	809.0	804.2	796.0	3.23	0.40
Slope (°)	2.91	16.03	2.32	0.28	2.38	81.81
Convexity (m)	0.02	2.69	0.02	−3.59	0.68	-

## Data Availability

The data are available upon request from the corresponding author.

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
