# Peer review of "Seasonal Influence of Biodiversity on Soil Respiration in a Temperate Forest"

_plants, 2022, doi:10.3390/plants11233391_

Round 1
Reviewer 1 Report
1. As we know, soil respiration (SR) in high latitude was limited in winter season. Do authors have any results in annual SR? It should be reported in MS for reader understanding the background in this study site.
2. Did authors measure soil respiration systematically or randomized? We also know soil temperature was increased in daytime and reached maximum afternoon. If authors systematically measured in three season, how to prevent systematically error in measurement.
Author Response
# Please see the revised manuscript in the attachment
- As we know, soil respiration (SR) in high latitude was limited in winter season. Do authors have any results in annual SR? It should be reported in MS for reader understanding the background in this study site.
We now provide annual total respiration for the study forest on lines 319-320 (Wu et al., 2006), and we also state clearly that we did not measure respiration in winter due to low temperatures constraining microbial activity (lines 359-360).
- Did authors measure soil respiration systematically or randomized? We also know soil temperature was increased in daytime and reached maximum afternoon. If authors systematically measured in three season, how to prevent systematically error in measurement.
There is a slight diurnal variation of soil respiration in forest ecosystems but it is very low in heavily shaded forested areas (ArchMiller et al., 2016; Davidson et al., 2000). It took two days to complete all measurements during each sampling campaign. We ensured that the influence of diurnal variation was minimized by performing the measurements in a random order between 10 a.m. and 4 p.m. on each measurement day (Jiang et al., 2020; Shi et al., 2016) Lines 356-359.
Wu, J.; Guan, D.; Wang, M.; Pei, T.; Han, S.; Jin, C., Year-round soil and ecosystem respiration in a temperate broad-leaved Korean Pine forest. Forest Ecology and Management 2006, 223, (1), 35-44.
ArchMiller, A. A.; Samuelson, L. J.; Li, Y., Spatial variability of soil respiration in a 64-year-old longleaf pine forest. Plant and Soil 2016, 403, (1-2), 419-435.
Davidson, E. A.; Verchot, L. V.; Cattânio, J. H.; Ackerman, I. L.; Carvalho, J. E. M., Effects of soil water content on soil respiration in forests and cattle pastures of eastern Amazonia. Biogeochemistry 2000, 48, (1), 53-69.
Jiang, Y.; Zhang, B.; Wang, W.; Li, B.; Wu, Z.; Chu, C., Topography and plant community structure contribute to spatial heterogeneity of soil respiration in a subtropical forest. Science of The Total Environment 2020, 733, 139287.
Shi, B.; Gao, W.; Cai, H.; Jin, G., Spatial variation of soil respiration is linked to the forest structure and soil parameters in an old-growth mixed broadleaved-Korean pine forest in northeastern China. Plant and Soil 2016, 400, (1), 263-274.

Reviewer 2 Report
The authors assessed the drivers of the spatial variability in soil respiration across spring, summer and autumn in a temperate forest in China. I read the paper with a lot of interest as the content reads well and the English language and style are fine. Only minor spell checks are required (see my minor comments below).
Regarding the content, I have just one major concern. I think that the use of the term "spatial variability in soil respiration" is not always justified, especially when the authors assessed the biotic and abiotic drivers of soil respiration. I agree with the authors when they used the CV as a metric of spatial variability in the section 2.1. However, in the subsequent analyses (i.e., regressions and SEM), the individual plot-level measure of soil respiration was used as the dependent variable instead of the CV, which means that the the models' outputs were based on the mean (and SE) soil respiration values. As such, I do not think that what is captured is the spatial variability. Moreover, the soil respiration was not measured at different spatial scales since all the measurements were done in a sampling unit of 20 x 20 m. Since the vegetation attributes were measured in 3 different radii around each sampling points (lines 420-421), I was expecting that the measurement of soil respiration could be done in these different radii so as to have some sort of different spatial scales. What is the opinion of the authors on that?
Here are additional minor concerns:
1- Given the format used for the presentation of the paper, specific biotic and abiotic parameters used should be indicated at the end of the introduction and acronyms should be defined to ease understanding of the subsequent result section.
2- Line 77: it would read better as: “….plant growth and microbial activities…..”
3- Line 84: should read “………..and is released back…….”
4- Line 439: change “form” to “from”
5- Line 464-467: could you provide a reference to support the thresholds used for the GOF statistics.
6- Lines 475-477: could you provide a reference to support the approach used to compute the relative contribution? Moreover, more information is needed to clarify these calculations. Which beta coefficient was considered? Direct or indirect or total beta value? Were both significant and non-significant beta coefficient values considered??
7- Lines 483: please replace “predations” by “predictions”
8- Lines 432-433: how was the slight positive spatial autocorrelation found in spring data dealt with?
Author Response
# Please see the revised manuscript in the attachment
- Regarding the content, I have just one major concern. I think that the use of the term "spatial variability in soil respiration" is not always justified, especially when the authors assessed the biotic and abiotic drivers of soil respiration. I agree with the authors when they used the CV as a metric of spatial variability in the section 2.1. However, in the subsequent analyses (i.e., regressions and SEM), the individual plot-level measure of soil respiration was used as the dependent variable instead of the CV, which means that the the models' outputs were based on the mean (and SE) soil respiration values. As such, I do not think that what is captured is the spatial variability. Moreover, the soil respiration was not measured at different spatial scales since all the measurements were done in a sampling unit of 20 x 20 m. Since the vegetation attributes were measured in 3 different radii around each sampling points (lines 420-421), I was expecting that the measurement of soil respiration could be done in these different radii so as to have some sort of different spatial scales. What is the opinion of the authors on that?
We agree entirely and apologize for the misunderstanding. The main aim of our study was to identify the drivers of soil respiration along local environmental gradients and to assess whether these drivers differ among seasons. We have now changed the title to better reflect the key aims and findings of our study and we have carefully checked the wording throughout the manuscript, replacing ‘spatial variation’ with ‘local-scale differences’ of ‘local-scale variability’ wherever appropriate.
Our study had two parts: 1) To demonstrate differences in soil respiration at local scales, we first quantified spatial variability using all sampling points. 2) We then investigated associations between biotic or abiotic variables and soil respiration to identify the mechanisms underpinning differences in soil respiration at local scales using the means for each sampling unit. This approach has been used in other studies (Jiang et al., 2020; Shi et al., 2016). Thus, we only selected the vegetation attributes in the most relevant radius (10-m radius; lines 432-434), rather than studying all three radii in detail.
Here are additional minor concerns:
1- Given the format used for the presentation of the paper, specific biotic and abiotic parameters used should be indicated at the end of the introduction and acronyms should be defined to ease understanding of the subsequent result section.
We have ensured all acronyms are explained at first mention in the text and have removed most acronyms from the results and discussion section to improve readability.
2- Line 77: it would read better as: “….plant growth and microbial activities…..”
Revised as suggested on line 81.
3- Line 84: should read “………..and is released back…….”
Revised as suggested on line 87.
4- Line 439: change “form” to “from”
Revised as suggested on line 449.
5- Line 464-467: could you provide a reference to support the thresholds used for the GOF statistics.
We now cite Rosseel 2012, which describes parameter threshold selection information (line 478).
6- Lines 475-477: could you provide a reference to support the approach used to compute the relative contribution? Moreover, more information is needed to clarify these calculations. Which beta coefficient was considered? Direct or indirect or total beta value? Were both significant and non-significant beta coefficient values considered??
We have removed the term “beta coefficient” to avoid confusion and replaced it with “total effect” (the sum of all direct and indirect effects for each predictor). We have defined the term and also provide a citation (Yuan 2019) on lines 486-488.
7- Lines 483: please replace “predations” by “predictions”
Revised as suggested on line 495.
8- Lines 432-433: how was the slight positive spatial autocorrelation found in spring data dealt with?
We assessed spatial autocorrelation to provide a comparative value of spatial variability in soil respiration (lines 242-243). However, the correlation was weak, and as the aim of our study was to explain local-scale differences in soil respiration, a low level of spatial autocorrelation is expected (Jiang et al., 2020).
Jiang, Y.; Zhang, B.; Wang, W.; Li, B.; Wu, Z.; Chu, C., Topography and plant community structure contribute to spatial heterogeneity of soil respiration in a subtropical forest. Science of The Total Environment 2020, 733, 139287.
Shi, B.; Gao, W.; Cai, H.; Jin, G., Spatial variation of soil respiration is linked to the forest structure and soil parameters in an old-growth mixed broadleaved-Korean pine forest in northeastern China. Plant and Soil 2016, 400, (1), 263-274.
Rosseel, Y., lavaan: An R Package for Structural Equation Modeling. Journal of Statistical Software 2012, 48, (2), 1 - 36.
Yuan, Z.; Ali, A.; Jucker, T.; Ruiz-Benito, P.; Wang, S.; Jiang, L.; Wang, X.; Lin, F.; Ye, J.; Hao, Z.; Loreau, M., Multiple abiotic and biotic pathways shape biomass demographic processes in temperate forests. Ecology 2019, 100, (5), e02650.
